# Effects of Enzymatic Hydrolysis on the Functional Properties, Antioxidant Activity and Protein Structure of Black Soldier Fly (*Hermetia illucens*) Protein

**DOI:** 10.3390/insects11120876

**Published:** 2020-12-09

**Authors:** Inayat Batish, Devon Brits, Pedro Valencia, Caio Miyai, Shamil Rafeeq, Yixiang Xu, Magdalini Galanopoulos, Edward Sismour, Reza Ovissipour

**Affiliations:** 1Department of Food Science and Technology, Virginia Tech, Blacksburg, VA 24060, USA; inayatbatish@vt.edu; 2Department of Entomology, Louisiana State University, Baton Rouge, LA 70803, USA; dbrits1@lsu.edu; 3Department of Chemical and Environmental Engineering, Technical University Federico Santa María, Valparaíso 2340000, Chile; pedro.valencia@usm.cl; 4Virginia Seafood Agricultural Research and Extension Center, Virginia Tech, Hampton, VA 23669, USA; caio_miyai@hotmail.com (C.M.); shamilrafeeq363@gmail.com (S.R.); 5Agricultural Research Station, Virginia State University, Petersburg, VA 23806, USA; yixu@vsu.edu (Y.X.); Galanopoulos@gmail.com (M.G.); esismour@vsu.edu (E.S.); 6Center for Coastal Studies (Coastal@VT), Virginia Tech, Blacksburg, VA 24061, USA

**Keywords:** insect proteins, black soldier fly, enzymatic hydrolysis, functional properties, antioxidant activity, amino acids composition, Raman spectroscopy, chemometrics

## Abstract

**Simple Summary:**

According to the FAO, the world’s population will reach 9 billion by 2050, and in order to provide enough food, meat production must increase by 100% and food production by 70%. Furthermore, more than 80% of fresh water resources are being used for agriculture, and 40% of the total food produced annually, is wasted. One sustainable agricultural practice involves converting by-products from the food and agriculture industry into valuable biomass, such as black soldier flies. Black soldier fly larvae can feed on by-products, and convert them to protein, carbohydrates, and oil. Black soldier flies could be used for feed and food development using different processing methods including enzymatic hydrolysis.

**Abstract:**

The effects of chemical protein extraction, and enzymatic hydrolysis with Alcalase, papain and pepsin, on the functional properties, antioxidant activity, amino acid composition and protein structure of black soldier fly (*H. illucens*) larval protein were examined. Alcalase hydrolysates had the highest degree of hydrolysis (*p <* 0.05), with the highest hydrolysate and oil fraction yield (*p* < 0.05). Pepsin hydrolysates showed the lowest oil holding capacity (*p* < 0.05), whereas no significant differences were observed among other enzymes and protein concentrates (*p* > 0.05). The emulsifying stability and foam capacity were significantly lower in protein hydrolysates than protein concentrate (*p* < 0.05). The antioxidant activity of protein hydrolysates from protein concentrate and Alcalase was higher than that with papain and pepsin (*p* < 0.05), owing to the higher hydrophobic amino acid content. Raman spectroscopy indicated structural changes in protein α-helices and β-sheets after enzymatic hydrolysis.

## 1. Introduction

The FAO has estimated that by 2050, the world’s population will reach 9 billion people, thus requiring food production to increase by 70% and meat production to increase 100% to meet global demands [1]. However, current agricultural practices are not sufficiently sustainable to address food insecurity concerns, and despite efforts to address this concern, one in eight people globally are food insecure, and one in six American children may not know where their next meal will come from. The importance of food security has been highlighted in the COVID-19 pandemic, during which many food processors and food supply chain stakeholders were shut down, thus creating a meat shortage and increasing food insecurity concerns. In addition, meat accounts for only 15% of the total energy in the global human diet, whereas approximately 80% of agricultural land is used for animal grazing or the production of livestock feed and fodder [2,3]. Meat consumption must be reduced by 70% to achieve sustainable food production systems and meet food security requirements [4]. Furthermore, food loss is another challenge in the sustainability, economics and the nutritional status of food. Despite considerable progress in agricultural production, post-harvest practices and supply chain management, approximately 30–40% of the total food produced is lost annually in the US [5]. Thus, there is an urgent need to develop novel and smart food production systems to reduce food waste, increase production yield and provide sustainable alternative proteins with minimum impact on the environment.

One sustainable food system is entomophagy, or eating insects, as a part of a diet widely followed in Asia, Africa and Latin America [6]. Insects represent the largest sector of fauna, accounting for 95% of biodiversity, and have historically been consumed at various stages of their life cycle [7]. In African countries such as Zambia and Nigeria, where the meat supply is insufficient, insects are a valuable source of protein [8]. Because of their sustainability, excellent nutritional value (protein 50–71%, fats 13.4–33.4% and fiber 5.1–13.6%), low emissions and greenhouse gas production, excellent feed conversion ratios, low water consumption and inexpensive feed sources, insects are a favorable candidate as alternative protein that may be developed for food and feed products [9].

The most important hurdle to the use of insects as food in Western countries is consumer acceptance. Studies have indicated that in developed countries, insects may be accepted by consumers when they are fragmented and included in a food as a protein powder or ingredient [10]. By applying enzymatic hydrolysis technology for protein recovery, a broad spectrum of food and feed ingredients might be produced with improved and upgraded functional properties and protein nutritional value [11].

From a food science and technology perspective, studies have developed protein hydrolysates from different insects including crickets (*Gryllodes sigillatus*) [12], migratory locusts (*Locusta migratoria* L.) [13], mealworms (*Tenebrio molitor*) [14] and black soldier flies (BSF) (*H. illucens*) [15,16,17,18]. Few studies have been conducted on BSF hydrolysis, including the antioxidant properties of BSF hydrolysates [16,18], and chemical and enzymatic hydrolysis of BSF for extraction and characterization of different fractions [15,17]. BSF enzymatic hydrolysis, functional properties, antioxidant activity, nutritional value and protein structure have not been evaluated through a systematic approach. This study aimed to process novel functional proteins from BSF larvae by using enzymatic hydrolysis to provide an alternative protein for developing food and feed.

## 2. Materials and Methods

### 2.1. One-Step Chemical Extraction of Protein

BSF larva meal was provided by Fluker Farms (Baton Rouge, LA, USA) and stored in a refrigerator until use in experiments (1 week). BSF meal was first defatted by mixing one-part BSF meal and two parts of petroleum ether (*w*/*v*) at room temperature in a shaking incubator for 1 h. The solvent containing lipids was removed, and the procedure was again repeated for the residues. Then lipids were recovered by solvent evaporation under a vacuum drier at 40 °C. The defatted pellet was washed three times with deionized water to remove solvent residue. The defatted BSF pellet (42 g) was treated with 40 mL of 1 M NaOH at 40 °C for 1 h. The supernatant was collected with a centrifuge, and protein was recovered by precipitation with 10% trichloroacetic acid solution in acetone. Samples were kept at −20 °C overnight, centrifuged at 4450× *g* for 30 min, and then washed three times with acetone and dried with a freeze drier to obtain BSF protein concentrate.

### 2.2. Enzymatic Hydrolysis of BSF Meal

The commercial proteolytic enzymes used in this study—Alcalase^®^, an endoprotease enzyme (2.4 AU/g) from *Bacillus licheniformis*; crude powder papain, a cysteine-protease from *Carica papaya* latex (1.5 units/mg); pepsin, an endoprotease from porcine gastric mucosa (250 units/mg)—were provided by Sigma-Aldrich Inc. (St. Louis, MO, USA).

The BSF larva meal (50 g) and distilled water (150 mL) (1:3 *w*/*v*) were mixed in a shaking incubator at room temperature for 2 h for hydration. Then the mixture temperature was increased to 60 °C with constant stirring in a mini shaker for 20 min at 220 rpm. Each enzyme was added at a ratio of 2% of the BSF meal. To enhance the enzymatic hydrolysis efficacy, 1% of each enzyme was added at the beginning of the process, and the other 1% was added after 60 min (two-step hydrolysis). Enzymatic hydrolysis was performed for 120 min. Hydrolysis was conducted at a pH of 6.85 for Alcalase and papain, and a pH of 3 for pepsin (pH was adjusted for pepsin using 0.1 M HCl). The enzyme for each mixture was inactivated at the end of the hydrolysis by heating the solution to 90 °C for 10 min in a water bath. The heated suspension was centrifuged (2500× *g*, 5 min, room temperature; Eppendorf 5417-R) thus resulting in three distinct phases including a semisolid phase at the bottom containing insoluble protein and chitin, an intermediate supernatant liquid phase containing protein hydrolysates and a light liquid phase at the top containing the lipid fraction. All samples were frozen at −20 °C, and the intermediate supernatant phase was separated and freeze-dried. The resulting freeze-dried protein hydrolysates were placed in sealed polystyrene conical tubes and stored at −20 °C until further use. The yield of hydrolysate fractions (wet weight basis) was determined according to the weight of the oil, hydrolysates and solid layers. All experiments were performed in six replicates (N = 6).

### 2.3. Degree of Hydrolysis

The degree of hydrolysis was measured according to formol titration as the proportion of α-amino N with respect to total N in the sample [19], in triplicate.

### 2.4. Analysis of Functional Properties

The functional properties of BSF protein concentrate and hydrolysates were determined in triplicate. The fat adsorption capacity of the BSF protein concentrate and hydrolysates was determined by mixing 500 mg of each sample with 10 mL canola oil [20,21]; this was followed by mixing and incubating samples for 30 min at room temperature with intermediate mixing every 10 min. The samples were then centrifuged at 2500× *g* for 30 min. Free oil was removed, and oil adsorption was evaluated according to weight differences and was expressed as ml of oil adsorbed by 1 g protein of BSF protein concentrate and hydrolysates.

Emulsifying stability (ES) was determined by placing 500 mg of each protein sample into a 250 mL beaker and mixing with 50 mL 0.1 M NaCl [22]. Then, 50 mL of pure canola oil was added to the mixture. A highspeed hand-held homogenizer was immersed in the mixture and operated for 2 min at maximum output to create an emulsion. From each emulsion, three 25 mL portions were placed in graduated cylinders and kept for 15 min at room temperature. Then the aqueous volume and total volume were measured. ES (%) was calculated as follows:(1)$$ES\;\left(\%\right)=\;\frac{Total\;volume\;-\;aqueous\;volume}{total\;volume}\;\times \;100$$

Foam capacity and foam stability were determined with the aeration method [23]. Briefly, 750 mg of protein sample was added to 25 mL of deionized water with a final pH of 6.8 and mixed with a stir bar for 10 min at room temperature. Protein mixtures were aerated with a homogenizer. Foam capacity was determined according to the percentage increase after aeration according to the following equation:(2)$$Foam\;capacity\;\left(\%\right)=\;\frac{volume\;after\;aeration\;-\;volume\;before\;aeration}{volume\;before\;aeration}\;\times \;100$$

Foam stability (%FS) was determined according to the percentage of foam remaining after 10, 30, 60 and 90 min.

### 2.5. Antioxidant Activity

The antioxidant activity of the hydrolysates was determined by 2,2,1-diphenyl-1-picrylhydrazyl (DPPH) free-radical scavenging assay. DPPH• solution was prepared by dissolving DPPH in 75% DMSO and dilution to a final concentration of 0.2 mM DPPH•. The hydrolysates (1.0 mL) were mixed with 1.0 mL of fresh DPPH• solution, incubated in the dark for 1 h, then measured at 515 nm with an Evolution 60 S spectrophotometer (Thermo Scientific, Pittsburgh, PA, USA) against a 75% DMSO blank. DPPH• radical scavenging activity was calculated with the following equation:(3)$$DPPH\;•\;radical\;scavenging\;activity\;\left(\%\right)\;=\;\left(1-\;\frac{As}{Ac}\right)\;*\;100$$
where *A_S_* is the sample absorbance, and *A_C_* is the absorbance of a blank control. The total protein in the hydrolysates was quantified with a colorimetric method with Thermo Scientific™ Pierce™ 660 nm protein assays. The samples were mixed with the assay reagent at the recommended 1:15 ratio. Pre-packaged, prediluted bovine serum albumin was used as the protein standard, with concentrations ranging from 125 μg/mL to 2000 μg/mL. Protein concentrations were measured with a NanoDrop 2000 spectrophotometer (Thermo Scientific, Pittsburgh, PA, USA) with reference to the absorbances obtained for a series of standard proteins.

### 2.6. Amino Acid Composition

The samples were hydrolyzed for 16 h at 130 °C in HCl (vapor phase), and this was followed by derivatization with Waters AccQTag derivatization reagents. Derivatized amino acids were quantified with RP UPLC, with a C18 analytical column (1.7 µm, 2.1 × 100 mm) and acetonitrile/water as buffers.

### 2.7. Raman Spectroscopy

Raman spectra were collected with a DXR2 microscopy Raman spectrometer (Thermo Fisher Scientific Inc., Waltham, MA, USA) equipped with a 785 nm diode laser source. Spectra were collected from 1700 to 1500 cm^−1^ (protein amide I and amide II regions) with a spectral resolution of 5 cm^−1^ under fixed parameters including a laser power of 20 mW, an average of five measurements and 100 scans. Spectra were collected from 10 mg/mL protein solutions.

### 2.8. Data Analysis

Each experiment was conducted with at least three replicates (*n* = 3) to ensure reproducibility. The results are expressed as the mean of the replicates ± standard deviation. The significance of differences among the biofilm removal treatments was determined with one-way analysis of variance, and differences were considered significant at *p* < 0.05. Raman spectra were pre-processed by using baseline correction, and this was followed by normalization and smoothing to flatten the baseline and remove noise. Second derivative transforms with the Savitzky–Golay filter with a gap value of 11 cm^−1^ were applied to reduce the spectral overlap and enhance discrimination of the spectral signature. Supervised chemometric models for the 1700–1500 cm^−^^1^ region of the protein concentrate and protein hydrolysate spectra associated with the amide I and II regions of proteins were developed to perform principal component analysis (PCA) and obtain loading plots, with Unscrambler^®^ X software (version 10.5) (CAMO Software, Oslo, Norway).

## 3. Results and Discussion

### 3.1. Degree of Hydrolysis and Yield

The degree of hydrolysis (DH) of BSF proteins hydrolyzed by Alcalase, papain and pepsin in a two-step enzymatic hydrolysis process in 120 min is presented in Table 1. The highest DH was achieved by Alcalase (18.4%), followed by papain (15.34%) and pepsin (9.8%), thus suggesting that Alcalase is the most suitable enzyme for the BSF protein hydrolysis process. These results are similar to those from previous studies on the enzymatic hydrolysis of migratory locusts (*Locusta migratoria*) [13], tropical banded crickets (*Gryllodes sigillatus*) [12], BSF (*H. illucens*) [18] and lesser mealworms (*Alphitobius diaperinus*, LM) [24]. In contrast, another study has shown that the lowest DH for BSF hydrolysates is associated with Alcalase (6%), as compared with papain (25%), pepsin (17%) and pancreatin (25%) [15]. This difference may be explained by two reasons: the difference in the enzyme to substrate ratio and a reduction in the enzymatic reaction rate due to limitation of the enzyme activity because of formation of inhibitory products, enzyme inhibition and enzyme deactivation [25,26]. Caligiani et al. [15] used a 1% enzyme to substrate ratio and hydrolyzed BSF for 24 h. However, in this study, we applied a 2% enzyme to substrate ratio in a two-step enzymatic hydrolysis to improve the enzymatic reaction and increase the degree of hydrolysis by adding 1% enzyme to substrate at the beginning and 1% after 60 min of enzymatic hydrolysis. Moreover, a strong relationship between the enzyme to substrate ratio and DH has been shown during enzymatic hydrolysis of tropical banded crickets (*Gryllodes sigillatus*) [12] and BSF (*H. illucens*) [16].

The yield of different hydrolysis fractions including BSF hydrolysate, oil and insoluble solid are presented in Table 1. The results illustrated that the highest hydrolysates and oil fraction yields were achieved by Alcalase, with the lowest solid fraction. In contrast, enzymatic hydrolysis with papain resulted in the lowest hydrolysate and oil fraction yield, with the highest amount of solid layer. Higher oil and hydrolysate recovery with Alcalase compared to Protamex and Flavourzyme during hydrolysis of sardines (*Sardina pilchardus*) [27], and Alcalase compared to several commercial enzymes during hydrolysis of anchovies (*Clupeonella engrauliformis*) [11] have been reported. Caligiani et al. [15] indicated that Alcalase, as compared with papain and pepsin, shows the highest protein yield during a 24 h hydrolysis process of BSF, with 10% oil recovery for all enzymes; this value is significantly higher than that in our study. This difference may be explained by the difference in BSF meal compositions and the longer enzymatic hydrolysis, which might have increased the oil recovery from the intact protein. The protein concentrate yield from a defatted BSF pellet was 24%.

### 3.2. Functional Properties

The results of oil adsorption are presented in Figure 1a–c as a function of hydrolytic enzymes. The oil adsorption of BSF protein concentrate and protein hydrolysate results indicated that the pepsin-hydrolysate oil adsorption was significantly lower than that of other protein hydrolysates and protein concentrate (*p* < 0.05), whereas there was no significant difference among Alcalase, papain and protein concentrate. These results indicated that pepsin did not enhance peptide functional properties, did not develop peptides with proper hydrophobic residues (CH_3_) and had significantly lower aromatic amino acid content (phenylalanine and histidine, tyrosine) [28]. The amino acid composition results from this study illustrated that the hydrophobic amino acid (HAA) content was significantly lower in pepsin hydrolysates than Alcalase and papain hydrolysates and protein concentrate (Table 2). The results from this study are consistent with those from another study that has reported the potential of Alcalase to produce the highest number of hydrophobic amino acids from white shrimp (*Litopenaeus vannamei*), thereby increasing the protein hydrolysate oil holding capacity [29]. Migratory locust (*Locusta migratoria* L.) protein hydrolysate oil adsorption results have shown improved oil adsorption with enzymatic hydrolysis with Neutrase, Flavourzyme and a mixture of enzymes, as compared with protein concentrate [13]; these results are in line with our findings in this study.

The ES of proteins can be defined as a protein’s ability to form and stabilize emulsions. The ability to form emulsions is an essential characteristic required by any protein-based moiety to be used in food-based applications, such as imparting functionality [30]. Enzymatic hydrolysis of BSF protein concentrate significantly decreased the emulsification properties from 100% to 40% in hydrolysates. Among the hydrolysates, papain had the lowest ES value (40%), which was significantly lower than those of Alcalase and pepsin. Only a few studies have demonstrated the emulsifying properties of insect protein [12,13]. Strong emulsification properties have been reported for intact insect protein including that from moth (*Cirina forda*) larvae [8], migratory locusts (*Locusta migratoria* L.) [13], *Gryllodes sigillatus, Schistocerca gregaria* and *Tenebrio molitor* [31]. Our study showed that the ES for protein concentrate was significantly higher than that of the hydrolysates, thus indicating that enzymatic modification of BSF larva protein did not improve the ES. A strong correlation between peptide size and emulsifying properties has been demonstrated. For example, salmon protein hydrolysates with a higher degree of hydrolysis show poorer emulsifying properties [20], a finding associated with larger peptides enhancing emulsifying properties [20]. In addition, the presence of some polysaccharides can elevate the viscosity and in turn, increase the stability of emulsions [31]. Zielińska et al. [31] also found a strong positive correlation between insect polysaccharide content and ES. In this study, we used a chemical method for extracting protein from BSF larvae according to previous studies [15]. Poor chitin extraction from BSF protein was observed with the chemical method, in contrast to a high chitin yield during enzymatic hydrolysis of BSF larvae [15]. In this study, we used both chemical and enzymatic methods for separating BSF larval protein. More research is needed to explain the emulsifying properties and potential links to chitin content, which may affect emulsifying properties.

The results of the foaming capacity (%) of BSF protein concentrate and protein hydrolysates illustrated that the highest foaming capacity was associated with BSF protein concentrate, with 20% foaming capacity, followed by Alcalase and papain hydrolysates, with 4% foaming capacity. The lowest foaming capacity was observed in proteins hydrolyzed with pepsin. Overall, all proteins showed unstable foam stability (<2 min). Poor to no foaming capacity and stability have been reported for several edible insects. For example, 6% foaming capacity has been reported for pallid emperor moth (*Cirina forda*) powder, which has low foam capacity and foam stability [8]. Improved foam capacity and foam stability after moderate enzymatic hydrolysis, as compared with that of insect powder, has been reported [12,24,31]. Intensive enzymatic hydrolysis results in a higher degree of hydrolysis and smaller peptides with poor foam capacity. Leni et al. [24] have shown that intensive enzymatic hydrolysis of protein from lesser mealworms decreases the foaming capacity, whereas protein hydrolysates with 5 to 10% DH, show 5 to 73% foaming capacity, and protein hydrolysates with 15% DH show no foaming capacity. In this study, we used two step enzymatic hydrolysis, which resulted in protein hydrolysates with high DH and poor foaming capacity. Moreover, these previous studies have compared the foaming capacity of protein hydrolysates with that of intact protein. However, the control group in our study was alkali extracted protein concentrate, which had less than 3% DH with high foaming capacity. Extracted protein from three edible insects, mealworms (*Tenebrio molitor*), tropical house crickets (*Gryllodes sigillatus*) and desert locusts (*Schistocerca gregaria*), has shown strong foaming capacity and foam stability, as compared with that of the intact insect protein [31]. Caligiani et al. [15] extracted protein from BSF with three methods and reported that one-step chemical protein extraction, as used in the current study, resulted in protein with a strong foaming capacity.

### 3.3. Amino Acid Composition

The amino acid compositions of BSF meal, BSF protein concentrate and BSF protein hydrolysates are presented in Table 2. The most dominant amino acid in the protein hydrolysates and intact protein was glutamic acid. Similar results have been reported by other researchers for enzymatic hydrolyzed BSF meal [15,16,18,32]. The most dominant amino acid in protein hydrolysates from other insect species such as tropical banded crickets (*Gryllodes sigillatus*) [12] and housefly larvae [33] is also glutamic acid. HAA (Ala, Ile, Leu, Phe, Pro, Tyr and Val), which are associated with peptides’ bioactive properties, such as antioxidant ability [34], were higher in BSF protein concentrate, followed by Alcalase and papain hydrolysates, and the lowest HAA values were associated with pepsin hydrolysates and intact protein. Similar results were observed for essential amino acids (EAA), positively and negatively charged amino acids (PCAA and NCAA) and aromatic amino acids (AAA). The results illustrate that protein concentration and enzymatic hydrolysis increased the quality and functional properties of the protein from BSF. However, depending on the enzymatic hydrolysis process and the type of enzyme, the functional properties and bioactivity of the peptides may be negatively affected. Other researchers have also indicated that enzymatic hydrolysis with Alcalase increases the quality of the amino acid composition of insect protein hydrolysates [12,16,18]. Leni et al. [24] compared the effects of different commercial enzymes on BSF protein hydrolysate free amino acid composition and found that pepsin and papain assisted hydrolysis result in the lowest and highest free amino acid content, respectively.

### 3.4. Antioxidant Activity

The ability of various protein hydrolysates to scavenge DPPH radicals is shown in Figure 2. All protein hydrolysates were able to scavenge of DPPH radicals, in accordance with other findings including those in housefly larvae [33], mealworm (*T. molitor*) larvae protein hydrolysate [14] and BSF (*H. illucens* L.) [18]. In this study, Alcalase hydrolysate showed higher antioxidant activity than protein concentrate and other hydrolysates produced by papain and pepsin. This finding may be explained by the higher HAA content in protein hydrolysates produced by Alcalase [35]. Wang et al. [33] have reported high antioxidant properties in housefly larvae protein hydrolysates, owing to the high amount of HAA. Zhu et al. [18] have also demonstrated that peptide fractions with more than 50% HAA have the highest antioxidant properties. An increase in the presence of peptides with hydrophobic amino acids located at the water–oil interface act as electron donors that augment the scavenging of DPPH radicals [36]. The results of amino acid composition analysis in this study showed significantly higher HAA content in protein concentrate and protein hydrolysates produced by Alcalase rather than papain and pepsin (Table 2). Higher antioxidant properties in hydrolysates produced by Alcalase have also been reported for other protein sources, such as anchovy sprat (*C. engrauliformis*) fish [11]. Hydrophobic amino acid residues can increase the presence of peptides at the water/lipid interface and therefore facilitate the scavenging of free radicals [11].

### 3.5. Raman Spectroscopy and Chemometrics

Figure 3 indicates a comparison among chemically (protein concentrate) and enzymatically (protein hydrolysates) extracted protein structures. The amide II (1500–1600 cm^−1^) and amide I (1600–1700 cm^−1^) regions provide secondary structural information about proteins, including C=O stretching, C–N stretching and N–H in plane bending of backbone peptide groups [37]. The amide II region showed several peaks including those at 1555 and 1587 cm^−1^ which were assigned to amide II and phenylalanine, respectively. The peak intensity decreased after enzymatic hydrolysis. Protein concentrates showed the highest peak intensity, followed by Alcalase, papain and pepsin hydrolysates. In the amide I region, several peaks were observed, among which 1645, 1655, 1658 and 1667 cm^−1^ were assigned to the α-helical structure of the amide I [38]. With enzymatic hydrolysis with Alcalase and papain, the peaks became more defined than those of protein concentrate, whereas weak bands were observed for pepsin protein hydrolysates, thus indicting strong protein denaturation. Reduced and absent α-helical structures in heat and acid treated protein have also been reported for fish protein, such as salmon [37], and for cricket microwave-assisted hydrolysis [39]. Bands around 1618 and 1622 cm^−1^ were assigned to tryptophan [38,39] and disappeared after enzymatic hydrolysis, which is in agreement with other research findings [39]. The peak intensity around 1622, 1676 and 1680 cm^−1^ significantly decreased during hydrolysis, thus indicating an absence of β-sheet structure and protein aggregation.

The PCA results illustrated that the spectral changes in the amide I and II regions of the BSF proteins were dependent on processing and enzymes, and the PCA model discriminated spectral changes in protein hydrolysates processed with different enzymes. In the PCA model for BSF proteins, the PC1 and PC2 components explained 47 and 36% of the variation, respectively, in the spectral band corresponding to the protein region.

Loading plots were prepared to identify the contributions of key wavenumbers to the PC1 and PC2 analysis. The key wavenumbers identified with loading plots can aid in understanding the biochemical and structural transformation induced in BSF protein after processing with different enzymes. The loading plot for BSF proteins illustrated that the major peak with the greatest contribution to differences in PC1 was around 1602 cm^−1^, thus illustrating phenylalanine changes and conformation in the amide I region. The other larger peaks were 1676, 1643, 1550 and 1583 cm^−1^, which were assigned to amide I (β-sheet), amide I (α-helix), tryptophan and the C=C bending mode of phenylalanine, respectively [38]. The major peaks with the greatest contribution to differences in PC2 were around 1594, 1533, 1603 and 1626 cm^−1^ from highest to lowest contributions. These wavenumbers have been assigned to phenylalanine amide carbonyl group vibrations and aromatic hydrogens, and the C=C in-plane bending mode of phenylalanine, tyrosine and tryptophan, respectively, in prior studies [38]. PC1 and PC2 indicated protein denaturation, aggregation and free amino acid formation as a result of enzymatic hydrolysis of BSF protein.

## 4. Conclusions

This study evaluated the effects of three commercial enzymes—Alcalase, papain and pepsin—on the degree of hydrolysis, protein and oil fraction yields, functional properties, antioxidant activity, amino acid composition and protein structure of BSF larvae. The results showed that, under two-step hydrolysis, Alcalase produced protein hydrolysates with a higher degree of hydrolysis, better functional properties, greater antioxidant activity and amino acid compositions with higher levels of HAA. Compared with the conventionally extracted protein (protein fraction), enzymatic hydrolysis reduced the functional properties in BSF hydrolysates; the lowest measured parameters were associated with the pepsin enzyme, mainly because of the poor amino acid composition of peptides. Enzymatic hydrolysis of BSF protein with Alcalase and papain offered a sustainable processing method, which may result in protein hydrolysates with a higher content of amino acids. The results from this study provide a baseline for developing sustainable alternative feed and foods from insects.

## Figures and Tables

**Figure 1 insects-11-00876-f001:**
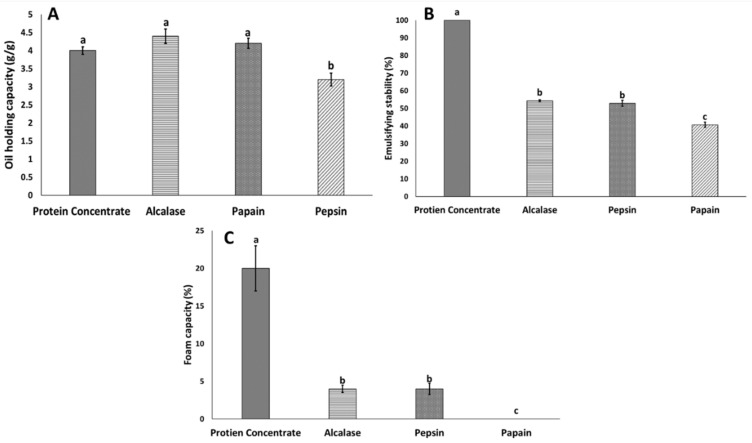
Functional properties of black soldier flies (BSF) protein concentrate and BSF hydrolysates. (**A**) Oil holding capacity; (**B**) emulsifying stability (ES); (**C**) foam capacity. Values with different letters are significantly different (*p* < 0.05).

**Figure 2 insects-11-00876-f002:**
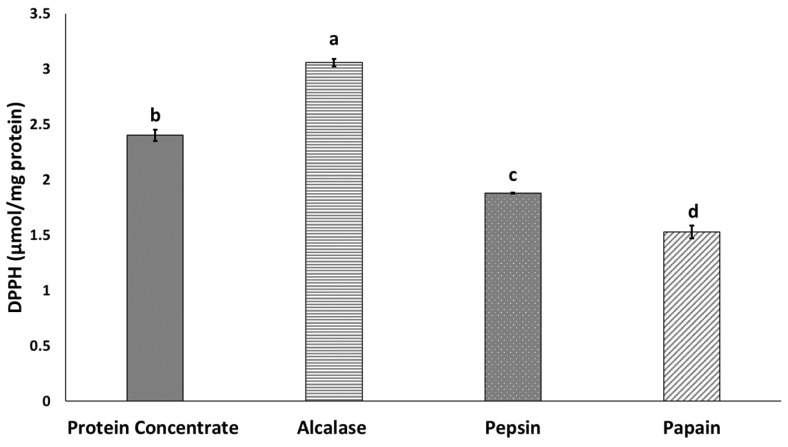
Antioxidant properties of BSF protein hydrolysates. Values with different letters are significantly different (*p* < 0.05).

**Figure 3 insects-11-00876-f003:**
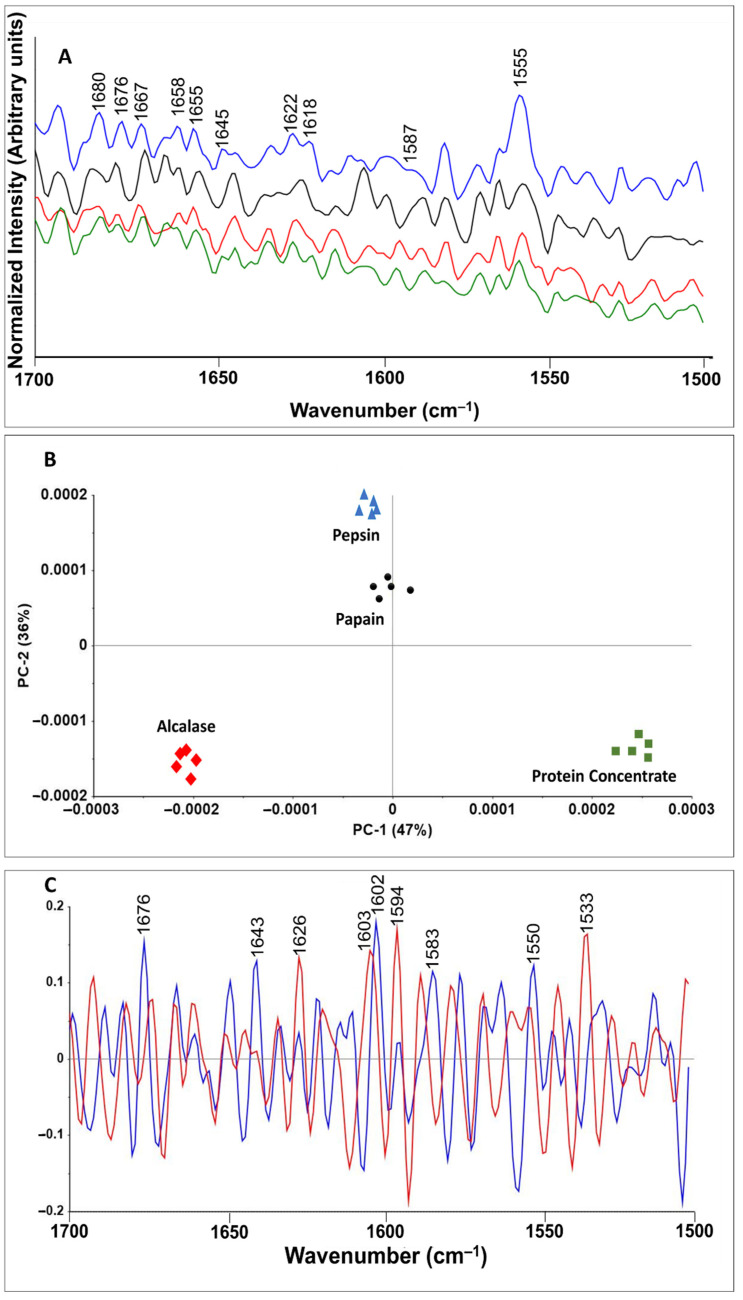
(**A**) Effect of enzymatic hydrolysis on Raman spectra (1700–1500 cm^−1^) of BSF protein: protein concentrate (blue); Alcalase hydrolysate (black); papain (green), pepsin (red); (**B**) principal component analysis (PCA) model of protein concentrate and protein hydrolysates; (**C**) loading plot of BSF proteins: PC1 (blue), PC2 (red).

**Table 1 insects-11-00876-t001:** Details of enzymes used, degree of hydrolysis and yield ^1^.

Enzyme	pH	Temperature (°C)	DH (%)	Yield (Wet Weight Basis%)		
				Hydrolysates	Oil	Solid Layer
Alcalase 2.4 L	6.85	60	18.4 ± 1.5 a	51.4 ± 1.7 a	7.1 ± 1.6 a	41.6 ± 2.9 a
Papain	6.85	60	15.34 ± 1.1 b	37.8 ± 1.1 b	4.6 ± 0.3 b	57.5 ± 1.1 b
Pepsin	3	37	9.8 ± 2.3 c	44 ± 3.2 c	3.2 ± 0.6 c	52.8 ± 3.8 c

^1^ Values are means ± SE (*n* = 6). Values in columns with different letters are significantly different (*p* < 0.05). DH: degree of hydrolysis.

**Table 2 insects-11-00876-t002:** Amino acid composition of BSF intact protein, protein isolate and hydrolysates ^1^.

Amino Acid	Quantity (mg/g)					
	Intact Protein	Protein Concentrate	Alcalase Hydrolysate	Pepsin Hydrolysates	Papain Hydrolysates	Reference Protein FAO/WHO 1985
ALA	10.95	38.94	41.90	11.95	34.71	
ARG	6.34	31.81	28.86	9.46	22.86	
ASP	13.97	82.42	58.51	18.71	46.29	
GLU	16.75	95.48	74.86	35.19	68.31	
GLY	7.82	31.67	33.15	11.61	32.16	
HIS	5.09	19.35	20.71	10.39	18.69	15
ILE	6.91	35.97	25.44	5.17	16.94	30
LEU	9.86	50.63	37.43	5.54	24.97	59
LYS	8.64	54.08	36.12	7.52	27.04	45
PHE	6.06	34.43	20.21	3.78	12.31	38
PRO	9.15	31.11	38.47	14.51	34.30	
SER	3.84	13.74	14.41	4.11	11.99	
THR	4.42	16.63	18.14	4.45	14.14	23
TYR	7.10	27.62	30.51	7.01	23.85	
VAL	9.79	43.40	38.60	7.75	30.31	39
HAA ^1^	59.83	262.11	232.56	55.70	177.37	
PCAA	20.06	105.23	85.68	27.37	68.59	
NCAA	30.72	177.90	133.37	53.91	114.60	
TEAA	50.78	254.49	196.65	44.60	144.39	
EAAI	0.3	1.1	0.85	0.27	0.7	
AAA	19.31	79.38	77.45	22.59	63.14	

^1^ HAA: hydrophobic amino acids; PCAA: positively charged amino acids; NCAA: negatively charged amino acids; TEAA: total essential amino acids; EAAI: essential amino acid index; AAA: aromatic amino acids.

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
