# Peer review of "Effects of Enzymatic Hydrolysis on the Functional Properties, Antioxidant Activity and Protein Structure of Black Soldier Fly (Hermetia illucens) Protein"

_insects, 2020, doi:10.3390/insects11120876_

Round 1

Reviewer 1 Report

85. How much the defatted BSF pellet was treated with 40 ml of 1 M NaOH at 40°C for 1 h?  

88. What number of fat-free protein concentrate was obtained from a number of raw biomass of the larvae? An important point in studying the process of processing biomass is the quantitative indicators of the yield of the target product, in this case, protein concentrate.

136. At the first mention in the text, you need to decipher the full name DPPH. It is a stable free radical. When interacting with antioxidants, it reduces the optical density. The method is quite old, but many people use it. Especially in the food industry. I think the chemiluminescent system is more sensitive than the absorption system. But the use of this method is quite possible.

196. An interesting result was obtained during the analysis the yield of different hydrolysis fractions are presented in Table 1. The sum of hydrolysates, oil, and solid layer was 101% for Alcalase, 99.9% for Papain, and 100% for Pepsin. What doesn't prevent you from getting values in columns with different letters are significantly different (P < 0.05). The methods do not specify how pH 3 is achieved in the reaction mass during pepsin fermentolysis.

215. Table 2 should be inserted immediately after the first reference to the data presented in it: «The amino acid composition results from this study illustrated that the hydrophobic amino acid (HAA) content was significantly lower in Pepsin hydrolysates than Alcalase and Papain hydrolysates and protein concentrate (Table 2)». Now it is not even in the discussion of the amino acid composition, but after the discussion of 3.3. Antioxidant activity. This makes it difficult to perceive the material.

The work is up-to-date and can be published. Comments are small and not fundamental. The only thing that raises doubts is the validity of the authors ' use of a large dose of the Aldolase enzyme (2%). This approach is unlikely to have any practical use. But only as a scientific study.

Reviewer 2 Report

Dear Authors, 

I review your Manuscript entitled "Effects of enzymatic hydrolysis on the functional properties, antioxidant activity and protein structure of black soldier fly (Hermetia illucens) protein". in the attached file you can find some corrections and suggestions. I think that the article can be published after be implemented in the results.

Kind regards
